# Imaging of electrically controlled van der Waals layer stacking in 1*T*-TaS₂

Corinna Burri [1,2], Nelson Hua [1], Dario Ferreira Sanchez [1], Wenxiang Hu[1,2], Henry G. Bell[1,2], Rok Venturini [1,3], Shih-Wen Huang[1], Aidan G. McConnell [1,2], Faris Dizdarević[1,2], Anže Mraz [3,4], Damjan Svetin[3], Benjamin Lipovšek[5], Marko Topič [5], Dimitrios Kazazis [1], Gabriel Aeppli[1,2,6], Daniel Grolimund [1], Yasin Ekinci [1], Dragan Mihailović [3,4,7] ✉ & Simon Gerber [1] ✉

Van der Waals materials exhibit a variety of states that can be switched with low power at low temperatures, offering a viable cryogenic 'flash memory' required for the classical control electronics for solid-state quantum information processing. In 1*T*-TaS₂, a non-volatile metallic 'hidden' state can be induced from an insulating equilibrium charge-density wave ground state using either optical or electrical pulses. Given that conventional memristors form localized, filamentary channels which support the current, a key question for design concerns the geometry of the conduction region in highly energy-efficient 1*T*-TaS₂ devices. Here, we report *in operando* micro-beam X-ray diffraction, fluorescence, and concurrent transport measurements, allowing us to spatially image the non-thermal hidden state induced by electrical switching of 1*T*-TaS₂. The results reveal a long-range ordered switching region that extends well below the electrodes, implying that the self-organized, collective growth of the hidden phase is driven by charge rearrangement and concomitant lattice strain. Our combination of techniques showcases the potential of non-destructive, three-dimensional X-ray imaging to study bulk switching in microscopic detail, exemplified here by electrical control of the charge-density wave state of a van der Waals material.

Switching between low and high resistance states via electrical excitation holds potential for analog and neuromorphic computing applications[1–3]. A key requirement for this is the ability to reversibly transition between two resistance states, which can occur through mechanisms such as the formation of a conductive filament, ferroelectric and magnetic tunneling junctions, or phase transitions between crystalline and amorphous states[3–6]. There is substantial research primarily focused on identifying new candidates for fast and energy-efficient memory devices and cryo-computing[7]. Van der Waals (vdW) materials, due to their layered structure, are well-suited not just

to explore novel states and phase transitions[8–11] but are also potentially useful for scalable electronic devices[12–14].

Among vdW materials, the transition metal dichalcogenide 1*T*-TaS₂ attracts attention for its correlated electron phenomena, including various charge-density wave (CDW) states[15,16], superconductivity under pressure or doping[17,18], and a putative quantum spin liquid phase[19,20]. At low temperatures it is believed to exhibit a Mott insulating state[17,21], characterized by commensurate (C) CDW order formed by star-shaped polaron domains that tessellate the layers below 150–180 K[21,22]. Moreover, it shows non-thermal, reversible switching to

[1]PSI Center for Photon Science, Paul Scherrer Institute, Villigen, PSI, Switzerland. [2]Laboratory for Solid State Physics and Quantum Center, ETH Zurich, Zurich, Switzerland. [3]Department of Complex Matter, Jozef Stefan Institute, Ljubljana, Slovenia. [4]CENN Nanocenter, Ljubljana, Slovenia. [5]Faculty for Electrical Engineering, University of Ljubljana, Ljubljana, Slovenia. [6]Institute of Physics, EPF Lausanne, Lausanne, Switzerland. [7]Faculty of Mathematics and Physics, University of Ljubljana, Ljubljana, Slovenia. ✉e-mail: dragan.mihailovic@ijs.si; simon.gerber@psi.ch

a metastable, metallic "hidden" (H) CDW phase upon application of ultrashort optical or electrical pulses. The lifetime of this HCDW state is short at elevated temperatures and stable below ≈40 K[23–27].

The microscopic origin of the non-thermal switching of 1T-TaS₂ has been intensely studied but is not fully understood[23,24,28–33]. It can be described in terms of a low-temperature free-energy landscape that features a global minimum (the Mott insulating equilibrium state) and multiple local minima separated by potential energy barriers[23,24,32,33]. Excitation with a laser or current pulse can drive the system into these metastable states.

Optical switching of 1T-TaS₂ with femtosecond laser pulses, exciting electrons homogeneously, has been studied with both quasi-static[30,34,35] and time-resolved[23,31,36–41] techniques. Surface-sensitive scanning tunneling microscopy (STM) shows formation of conducting domain walls[23,30], while bulk-sensitive X-ray diffraction (XRD) reveals changes in the out-of-plane polaron stacking and a different in-plane commensurability of the HCDW state compared to the low- and room-temperature CDW states, confirming a non-thermal switching mechanism[31].

In contrast, electrical switching has been less explored despite its promise for application in highly-efficient cryo-memory devices due to the compatibility with standard electronics. The electrically-induced hidden (e-HCDW) state has been studied using transport measurements[25,26,42,43], STM[27–29], and angle-resolved photoemission spectroscopy[44], revealing macroscopic behavior similar to the optically-induced hidden (o-HCDW) state. Domain wall formation at the surface has also been observed with STM, but unlike the o-HCDW, it is understood that the e-HCDW state results from excitation due to charge carrier separation, leading to more inhomogeneous domain walls[27,32]. Furthermore, in contrast to the o-HCDW state, the e-HCDW can be triggered with much longer pulses up to 100 ms[45], suggesting key differences in their switching mechanisms. As previously discussed[27,32], the switching is non-thermal even for long current pulses.

It is unclear whether optical and electrical excitations lead to the same local free-energy minimum. If they do, can these two distinct perturbations—differing in pulse duration, electron-hole pairs accessed, and directionality—follow the same non-thermal pathway, or are there different mechanisms at play? The lack of structural studies on the e-HCDW also leaves the question unanswered whether the switching is surface-localized or propagates through the bulk. Likewise, it is unknown if the e-HCDW spreads across the entire region between the electrodes or forms conducting filament channels. Finally, it is not established whether the principal actor of the switching is the electric field or the current. Resolving the spatial extent of the e-HCDW would provide insights into the switching mechanisms, and thereby contribute to designing devices with improved performance and scalability.

The required information can be obtained by mapping the bulk CDW states before and after switching. If the e-HCDW is structurally equivalent to the o-HCDW, then the known CDW diffraction peaks from photo-driven experiments[31] can be used as a fingerprint to identify the e-HCDW switching region. Here we report on in situ resistance measurements with micro-beam X-ray fluorescence (μXRF) and diffraction (μXRD) to spatially image a nano-fabricated 1T-TaS₂ device *in operando* (Fig. 1)[46]. Analysis of CDW Bragg peaks reveals the detailed three-dimensional (3D) spatial evolution of the material properties before and after electrical excitation.

## Results

Figure 1b shows an optical image of the 1T-TaS₂ device, formed by a contacted flake, used to investigate the e-HCDW state (fabrication details in "Methods"). The distance between the electrodes was set to ≈8 μm, considering the μm-sized X-ray spot and that the switching

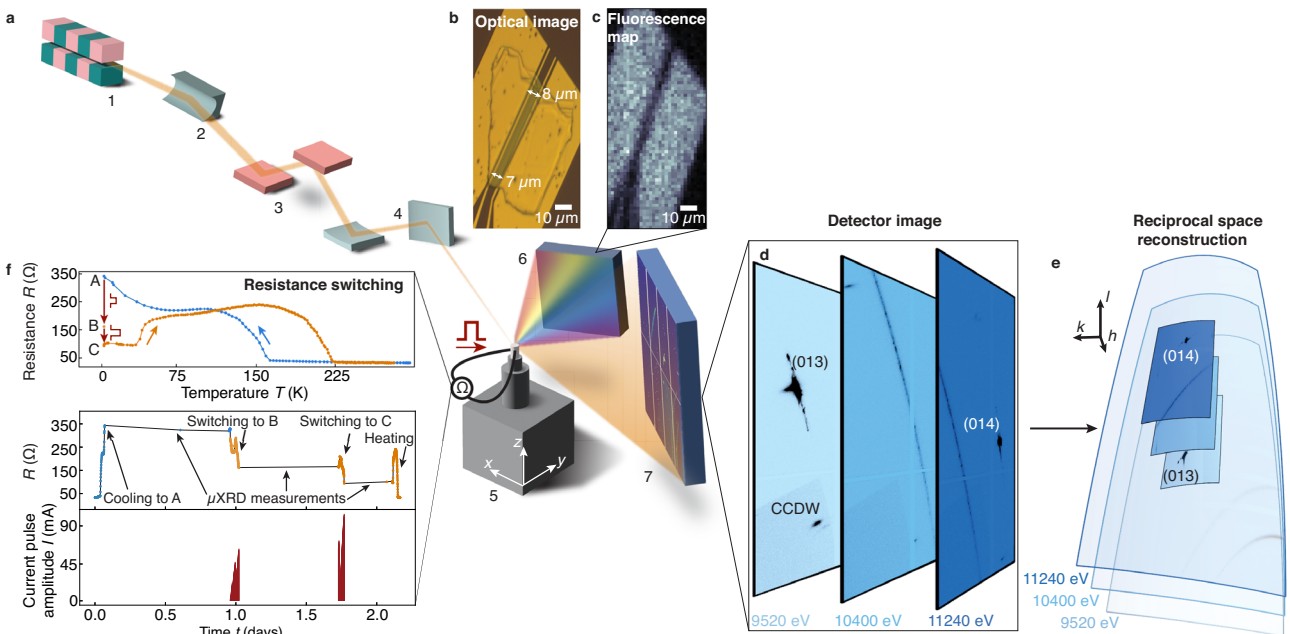

**Fig. 1 | *In operando* macro- and microscopic measurement of phase switching.** **a** Schematic of the synchrotron beamline including the undulator (1), a toroidal mirror (2), the monochromator (3) and Kirkpatrick-Baez focusing mirrors (4). The 1.5 × 2.5 μm² sized X-ray beam is directed at the 1T-TaS₂ device (5) in a ⁴He cryostat that can be moved using translation stages. The device is electrically contacted, allowing for resistance measurements and application of current pulses. X-ray fluorescence (6) and diffraction (μXRD) (7) is recorded simultaneously on respective detectors. **b** Optical image of the 1T-TaS₂ device. **c** Au fluorescence map highlighting the electrodes. **d** Diffraction patterns measured at 6 K at selected X-ray energies with the lattice (013) and (014) Bragg reflections, as well as one commensurate charge-density wave (CCDW) peak. **e** Conversion of the 2D detector images taken at various X-ray energies to 3D (hkl) reciprocal space. **f** In situ resistance upon cooling (blue) and heating (orange), and as a function of time with A the unswitched CCDW state, as well as B and C the partially- and fully-switched HCDW states, respectively.

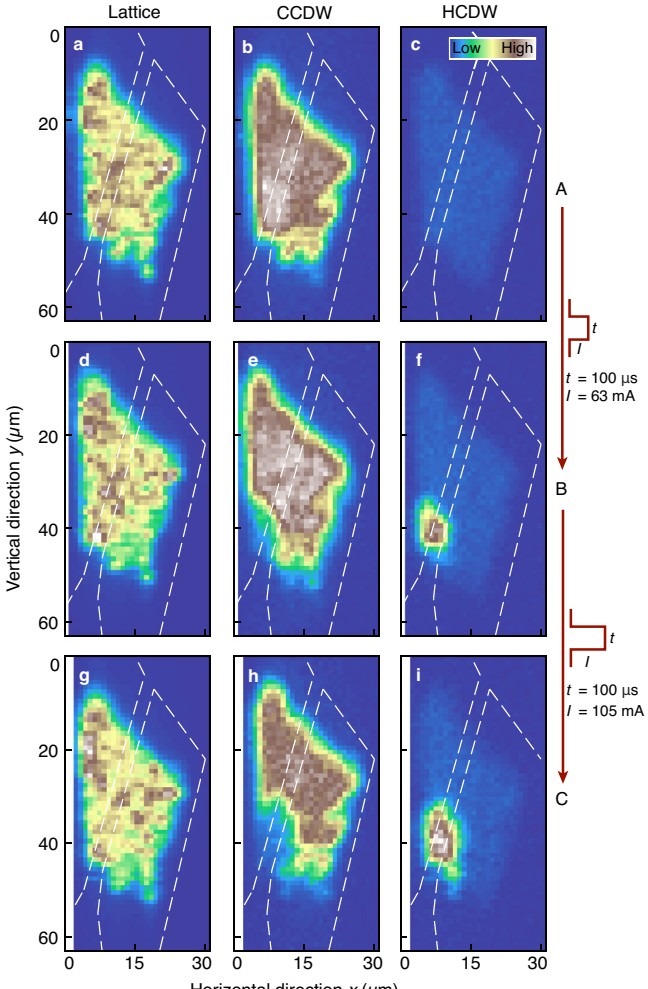

**Fig. 2 | Electrically-induced non-thermal phase switching. a–c** Spatially-resolved intensity at the reciprocal space positions of the (014) lattice reflection, as well as a nearby commensurate charge-density wave (CCDW) and hidden state (HCDW) peak in the unswitched state A. The lattice and CCDW peaks are observed across the whole flake, whereas no HCDW signal is found. **d–f** and **g–i** show the respective peak intensities in the partially- B and fully-switched state C. The CCDW signal is suppressed in the lower left corner of the flake where the HCDW signal emerges. Color scales show a minimal mean intensity of 0 counts and maxima of 1.50 (lattice), $5 \cdot 10^{-2}$ (CCDW), and $4 \cdot 10^{-2}$ (HCDW) counts for an integration time of 100 ms. Dashed lines indicate the position of the electrodes.

voltage scales with the distance between the electrodes[42]. There are two inert contacts between the outer electrodes that are not used for the resistance measurement. The device is located using $\mu$XRF (Fig. 1c) which is sensitive to even small traces of embedded elements[47,48] (Supplementary Methods). Before and after applying current pulses, $\mu$XRD and $\mu$XRF measurements are taken at room temperature and 6 K, as well as different energies spanning an extended portion of reciprocal space (Fig. 1d, e). During the $\mu$XRD and $\mu$XRF measurements the device is shorted and grounded. Conversion to 3D reciprocal space allows us to identify the equilibrium and newly appearing signals, as well as to determine their position and peak shapes (Supplementary Methods). Temperature-dependent resistance of the device is measured in a two-probe configuration (Fig. 1f). Upon cooling, the flake goes through a first-order phase transition, resulting in a step-like increase of the resistance at ≈150 K. We denote the unswitched low-temperature CCDW state with A. To induce the e-HCDW state, single square current pulses with a width of 100 μs are applied. The pulse

amplitude is increased gradually until a resistance drop occurs. We denote the state B with 47% resistance compared to A as partially-switched, reached by applying a pulse amplitude of 63 mA. Increasing the pulse height to 105 mA induces another resistance drop to 27% of A, to which we refer as fully-switched C. Following the resistance upon heating, we observe the characteristic relaxation from the HCDW state to the equilibrium high-temperature states with an intermediate resistance plateau commonly assigned to a partial relaxation of domains[24,32], as well as the first-order, hysteretic transition, confirming that the HCDW state is induced in the device. As a function of time, the device resistance exhibits changes upon cooling, heating, and current pulse application, else it is stable over hours. During switching, we observe some resistance increase, likely due to a partial relaxation to the CCDW state, and then, as the current amplitude is further increased, a resistance drop indicating that the HCDW state is induced. We note that devices with an optimized design can be switched with a single current pulse[42,43], whereas with our device design, optimized for in situ $\mu$XRD measurements, multiple pulses with increasing amplitude have to be applied. During the $\mu$XRD measurements at a constant temperature of 6 K the resistance remains stable. Importantly, the resistances measured at room temperature before and after the switching and $\mu$XRD measurements are identical, which directly proves the non-destructive nature of our technique.

Several structural changes occur in the non-equilibrium o-HCDW state compared to the equilibrium CCDW order: (i) the CCDW and "dimer" peaks vanish[31] (the latter are associated with inter-layer dimerization of the star-like domains), (ii) a new long-range order (at different positions in reciprocal space) appears as HCDW peaks[30,31], and (iii) the out-of-plane lattice constant contracts[49] (observed as a shift of the lattice peaks in the out-of-plane direction). In the following, we show that these three characteristic features are also observed for the e-HCDW state. Here, we focus on the HCDW signal, whereas the vanishing dimer CCDW peaks and the out-of-plane lattice contraction are addressed in the Supplementary Discussion.

For each position on the device, we reconstruct the 3D reciprocal space and look for a decrease in intensity of the CCDW peak, as well as newly appearing signals hinting at the e-HCDW state: three 3D regions of interest (ROI) around the (014) lattice, as well as nearby CCDW and o-HCDW positions[31] are set, and the intensity within these ROIs is integrated and averaged. There is more spatial variation in the lattice than the CCDW images, since the latter Bragg peak is broader (particularly along the out-of-plane direction) and the conversion to 3D reciprocal space is limited by the energy step size. The shape of the flake can be clearly seen in the lattice peak maps (Fig. 2a, d, g), which remains intact throughout the switching process. However, a change occurs in the CDW structure of the flake: the CCDW peak map before switching (A, Fig. 2b) shows the same outline as the lattice peak; but Fig. 2e, h in the partially- and fully-switched states, B and C, respectively, reveal that the CCDW peak intensity in the bottom left part of the flake vanishes. Concurrently, an e-HCDW peak signal appears in the same region between the electrodes (Fig. 2f, i). That is, not the entire flake switches from the CCDW to the HCDW order, but rather a conducting region appears between the electrodes on one edge of the flake, while the rest of the device remains in the equilibrium state.

Having identified the spatial region of the flake that switches to the HCDW state, we examine the momentum-space structure and real-space distribution of the CDW states. The CCDW signal in Fig. 3a features the characteristic elongation parallel to the out-of-plane $l$ direction due to partial disorder[50], and the reciprocal space position is close to the previously reported values[31,51,52] (Table 1). Figure 3b–d are from the fully-switched state C, deduced from different regions on the flake and revealing that CCDW order gradually gives way to the e-HCDW state as the switching region is approached. The reciprocal space position of the e-HCDW peak between the electrodes is also found to be consistent with the o-HCDW signal in the literature[31] (Table 1).

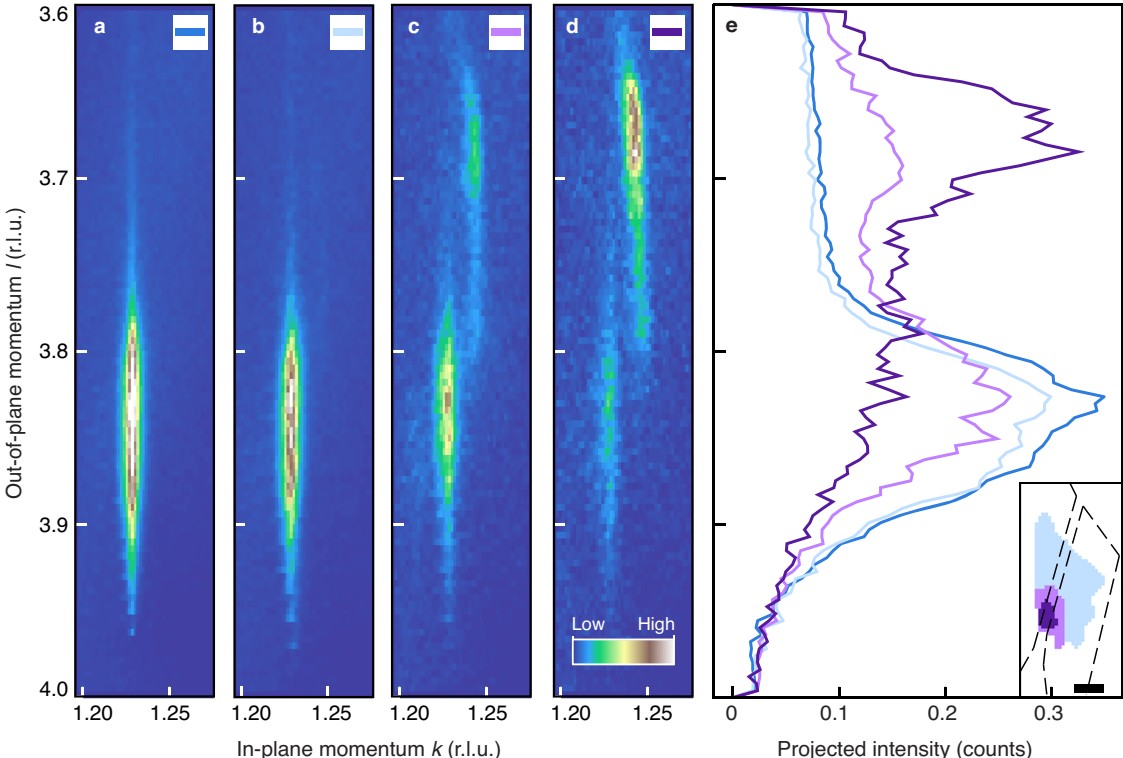

**Fig. 3 | Momentum- and real-space structure of the CDW states. a** 2D (*kl*) projection in reciprocal lattice units (r.l.u.) of the commensurate charge-density wave (CCDW) peak in the unswitched state A (dark blue), obtained by integrating the intensity along the in-plane *h* direction and normalizing per pixels. **b** Measurement of the fully-switched state C in the light blue region (inset of **e**), showing a CCDW and only a faint hidden state (HCDW) signal. **c, d** Respective projection measured in the light and dark purple regions (inset of **e**), close to and in the vicinity of the electrode gap, respectively. Fewer pixels in those regions result in poorer statistics of the projections. **e** Out-of-plane projection of the averaged intensities shown in (**a**–**d**) for an integration time of 100 ms. Dashed lines on the spatial map in the inset indicate the location of the electrodes. The scale bar is 20 μm.

**Table 1 | Peak positions from electrically- and optically-switched experiments**

| Peak | Literature | Experiment | Exp. uncertainty |
|------|-----------|-----------|-----------------|
| Lattice | (0, 1, 4) | (0.00, 1.00, 4.00) | (−5.2e⁻⁴, 1.4e⁻³, 4.9e⁻³) |
| CCDW | (0.08, 1.23, 3.80) | (0.08, 1.23, 3.83) | (1.0e⁻³, 7.7e⁻⁴, 3.4e⁻²) |
| HCDW | (0.07, 1.24, 3.67) | (0.07, 1.24, 3.68) | (3.5e⁻⁴, 5.8e⁻⁴, 4.2e⁻²) |

Reciprocal space coordinates (*hkl*) in reciprocal lattice units of the measured structural (014), commensurate charge-density wave (CCDW) and electrically-induced hidden state (HCDW) peaks, as well as the respective literature values for the CCDW[52] and the optically-induced HCDW[31] signals. The experimental position and uncertainty are determined from fitting Lorentzian peak shapes after the reciprocal space reconstruction.

Therefore, we conclude that the optically- and electrically-driven HCDW order is equivalent not only from an electronic but also a 3D structural point of view. In turn, this also implies control of the switching into the same local minimum of the free-energy landscape regardless of the excitation method.

Next, we use the intensity shift from the CCDW to the HCDW peak to assess the switching depth. We take the ratio of the HCDW and the total (HCDW + CCDW) signals as a proxy for the volume fraction of the switched layers. Since the Pd/Au electrodes are on top of the flake— thus, the highest current density is there—we also assume that the switching starts from the top. Figure 4 shows the respective 3D tomographic representation of the device. Clearly, the switching is not restricted to the surface but penetrates deep into the bulk of the 500-nm-thick flake. This observation is in agreement with previous reports on the importance of the out-of-plane layer reconfiguration in the HCDW state[29,31,50]. We take cuts parallel and perpendicular to the electrode gap to further characterize the bulk material switching: The

HCDW region extends in volume going from the partially- B to the fully-switched state C. By "partial" (63 mA) we mean that only a fraction of the "fully" switched volume at the maximal current of 105 mA is converted, but this portion switches completely from an electronic point of view. Parallel to the gap the cut reveals that the switching starts at the edge of the flake that is the shortest path to ground. Moreover, from the cut perpendicular to the gap we see that the electrons flowing from the − to the + electrode do not symmetrically switch the intergap space. A fraction of ≈17% and 9% remains unswitched at the bottom of the flake between the electrodes in the partially- and fully-switched state, respectively. Though using XRD, we cannot extract information down to the single layer and consequently the remaining unswitched intergap layers in the fully-switched state denote an upper bound. We also observe sizeable HCDW order induced under the electrodes that is not directly the shortest path to ground, hinting that not only the electronically-driven out-of-plane layer reordering[29,31,50], but also the resulting strain and change of the out-of-plane conductivity[45,53] play a role in the manifestation of the switching process.

## Discussion

We non-destructively acquire *in operando* "tomograms" of cryo-memory device switching, addressing a long-standing challenge for the engineering of phase-change memory devices[3,54]. Our approach allows us to identify the switched HCDW state along one edge of the flake and extending under the electrodes into the bulk of the material. This hints at charge injection as the initiator of the switching. The appearance of a current-induced conducting boundary at the flake edge/electrode interface also aligns with previous findings[55].

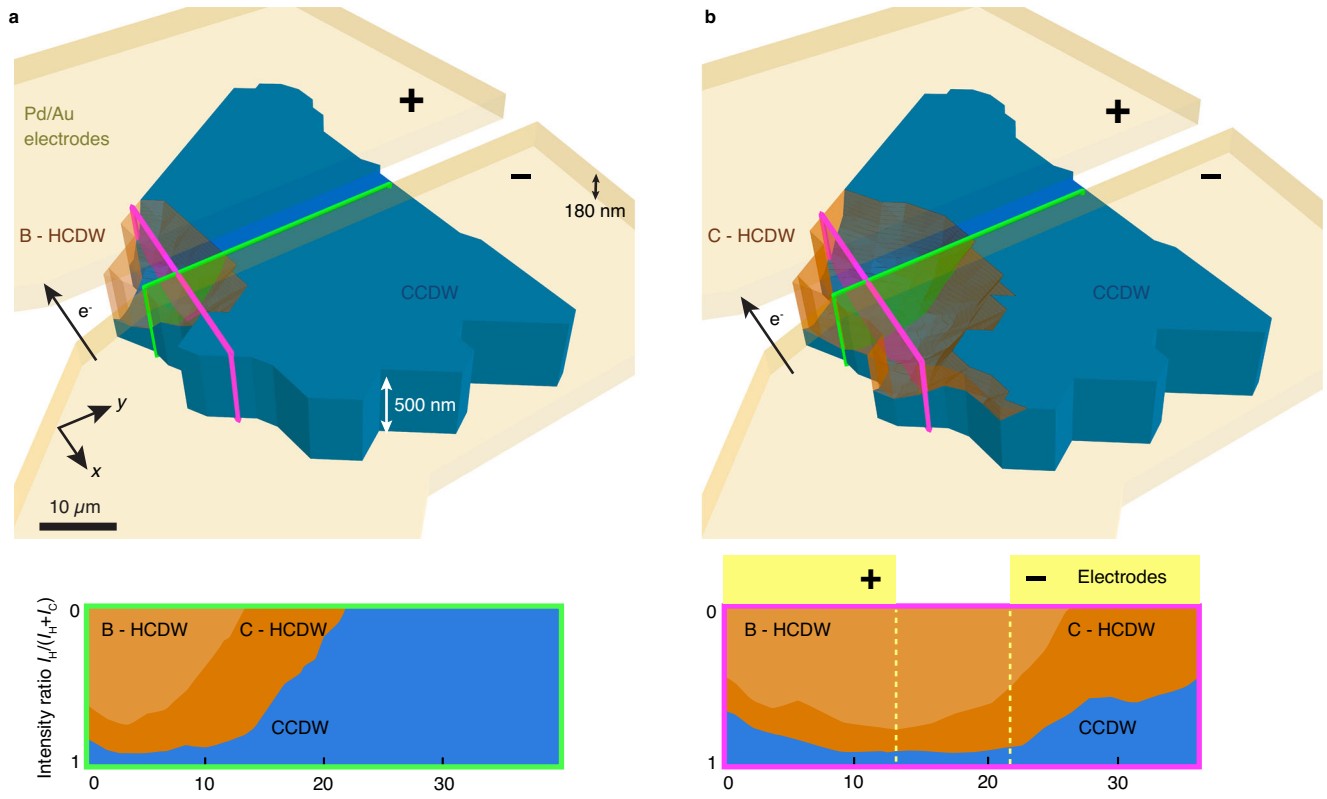

**Fig. 4 | Bulk electrical switching.** 3D tomographic representation of the device in **a** the partially- B and **b** the fully-switched state C. The vertical dimension represents the normalized switching depth, defined by the ratio of the hidden charge-density wave (HCDW) signal, as well as the total commensurate (CCDW) and hidden state intensity. 180-nm-thick Pd/Au electrodes (yellow) are on top of the 500-nm thick 1$T$-TaS$_2$ flake, where the CCDW and HCDW regions are depicted in blue and orange, respectively. Width and depth are not to scale. The arrow indicates the electron (e$^-$) flow upon application of the current pulse. Cuts through the flake parallel (green) and perpendicular (pink) to the electrode gap are shown on the bottom left and right, respectively. Going from B (light orange) to C (dark orange), the in-gap HCDW order extends both laterally towards the − electrode and in volume.

Finite element method simulations (Supplementary Discussion) indicate that the geometry of the device dictates the switching region's location: It occurs where the electrode gap is narrowest, providing the shortest path to ground and resulting in the highest current density during pulse application. Once this initial transformation has taken place, the current primarily flows through it, and the switched volume only extends when the current is further increased, e.g., from state B to C (Fig. 4). The Au electrodes ensure high conductivity along the flake, rendering a conductivity gradient unlikely[56]. Although the switching region exceeds the sizes of defects[57] and CDW domains[30], it cannot be ruled out that the switching nucleates there, or that current crowding effects at the interface between the electrode and the flake play a role. The lattice peak maps in Fig. 2a, d, g confirm that the flake remains intact during the switching process, ruling out structural fracture as the root cause, though the irregular shape of the flake could also favor a switching location.

The switching region can also be located by mapping the vanishing dimer peak (Supplementary Discussion), associated with the breaking of the pairwise alignment of vdW layers[31]. The nature of the equilibrium CCDW state—whether it is a Mott or a band insulator—remains controversial and is found to be tied to the out-of-plane restacking[50]. Here, we confirm control and local modification of the out-of-plane stacking via current pulses.

In- and out-of-plane strain between the flake and the substrate or electrodes may not only influence the stability of the HCDW state[24,58] but also the switching region's location. Studies on other phase-change materials show that local strain inhomogeneities can arise upon electrical triggering[59]. We also observe an out-of-plane lattice contraction

localized exclusively in the intergap space of our 1$T$-TaS$_2$ device when a current pulse is applied (Supplementary Discussion). This creates local strain between the switched and unswitched areas. Thus, we identify strain propagation via the CDW domains as an essential factor contributing to the localized and bulk nature of the switching, in close analogy to what occurs for ferroelectric devices[60]. Since strain is a long-range effect, it can also explain why the switching extends beneath the electrodes—an aspect not captured by our strain-free simulations.

Interestingly, the original CCDW intensity is higher in the switching region. This is not apparent for the lattice peak, which features stronger spatial variations that may obscure such correlations (Fig. 2a, b). Conversely, this could suggest that structural factors such as flake thickness are secondary and, instead, point to switching being promoted by a particularly coherent (well-ordered) CDW state. Our results, therefore, underscore the importance of resolving electronic heterogeneities and looking into the role they play in the non-thermal switching process of 1$T$-TaS$_2$.

Our 3D reconstruction of the phase change suggests optimized device designs using narrow flakes with electrodes positioned along the edges. Ultimately, we envision non-parallel separate electrode pairs on both flake edges, as well as orthogonal ones on top and bottom of the flake that allow for multiple switched regions and controlled cross talk between them, eventually enabling highly-efficient and fast logic operations.

Studying bulk switching using XRD with nano/micro-beams offers many opportunities. Direct imaging, e.g., of filamentary paths arising from strain or different structural reconfiguration, can also be applied to other memristive materials. So far, these have mostly been

investigated in thin films without contacts[61], in-plane with transmission electron microscopy, or indirectly and destructively using conductive atomic force microtomography[62–65]. Using a non-destructive, 3D, *in operando* approach provides microscopic detail of the switching mechanism, thereby advancing the design of scalable next-generation memory technologies.

## Methods

### Device preparation

1$T$-TaS$_2$ single-crystals were synthesized using chemical vapor transport with iodine as the transport agent leading to consistent crystal quality[19]. Flakes were mechanically exfoliated from bulk crystals using GelPak and excess flakes removed using Scotch tape such that a large, single flake could be deposited onto an oxidized Si wafer (oxide thickness: 280 nm) with predefined Ti/Au alignment markers. The electrodes were written on the flake using electron beam lithography, followed by physical vapor deposition of 20 nm Pd and 160 nm Au. The investigated flake has lateral dimensions of about $50 \times 50\,\mu m^2$ and a thickness of 500 nm determined using an optical microscope and a profilometer, respectively. The in-plane crystal orientation of the device was characterized at the Material Science beamline of Swiss Light Source synchrotron[66], and it was then glued onto a Cu sample holder using GE varnish.

### Transport measurements and sample cooling

The device was mounted in a CryoVac $^4$He cryostat with Kapton windows for the incident and scattered X-rays. The resistance was measured in a two-probe configuration on the flake, and converted to four-probes on the sample holder such that only the resistance of the Pd/Au electrodes on the flake was measured in series with the flake. The electrical setup consisted of a Keithley 6221 pulsed current source and a Keithley 2182 nano-voltmeter connected in delta mode. For electrical switching single, square-wave pulses with pulse lengths of 100 μs were applied using the current source. The current pulse amplitude was gradually increased until the resistance dropped. For the switching from B to C, we started with the current pulse amplitude where switching from A to B had occurred. But as we observed already during the partial switching, an intermediate increase in resistance occurred and we again applied lower pulses until the final decrease in resistance was measured. The sample temperature was monitored using a Cernox thermometer mounted on the sample holder about 2 cm from the device. A specific cooling procedure ensured that the first-order phase transition to the CCDW state was crossed in a controlled fashion. The following ramp rates were used: 300–250 K: 5 K min$^{-1}$, 250–200 K: 2 K min$^{-1}$, 200–100 K: 1 K min$^{-1}$, and 100–6 K: 5 K min$^{-1}$.

### X-ray diffraction and fluorescence imaging

Spatially-resolved X-ray scattering was performed at the microXAS beamline of the Swiss Light Source[67], allowing for simultaneous in situ resistance measurements, as well as μXRD and μXRF imaging. We performed two measurement rounds, observing qualitatively the same behavior across several devices. The incidence angle of the X-rays with respect to the device was fixed to ≈25°, resulting in a beam spot size ≈1.5 × 2.5 μm$^2$ (vertical × horizontal) using Kirkpatrick-Baez focusing mirrors. The nominal X-ray flux was adjusted to ≈10$^9$ photons/s. Considering that the X-rays were passing through ≈10 cm air before impinging on the sample, the effective incident flux on the sample was about 15% and 10% lower at 9.1 and 12.0 keV, respectively. As the cryostat was mounted on translation stages without rotational degree of freedom, to take μXRD spatial maps we scanned through reciprocal space by changing the incoming X-ray energy between 9.1 and 12.0 keV, with step sizes ranging between 20 and 80 eV, using the undulator and monochromator settings. The diffraction and fluorescence signals were recorded with an integration time of 100 ms by Eiger X 4M and VIAMP-KC detectors, respectively. To locate the device

the μXRF signal at the Au (11.9 keV) and Ta (9.88 keV) $L$-edges was used (Supplementary Methods).

## Data availability

The processed XRD data, already mapped to reciprocal space, have been deposited in the PSI Public Data Repository available at https://doi.org/10.16907%2Fb555e793-e64c-43a4-8dd7-7e66e37300f0.

## Code availability

Code developed for this study, including the determination of experimental constraints and reciprocal space reconstruction, gold ring detection and removal, as well as image alignment and plotting of the data have been deposited in the PSI Public Data Repository available at https://doi.org/10.16907%2Fb555e793-e64c-43a4-8dd7-7e66e37300f0.

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

## Acknowledgements

The authors acknowledge the Paul Scherrer Institute, Villigen, Switzerland for provision of synchrotron radiation at the microXAS and Material Science beamlines of the Swiss Light Source. We thank P. Sutar for synthesizing the 1$T$-TaS$_2$ crystals and acknowledge technical support from the PSI PICO operations team. Furthermore, we are grateful to B. Meyer, M. Birri, and S. Stutz for technical support before and during the beamtimes, as well as fruitful discussions with N. Taufertshöfer. This research was funded by the Swiss National Science Foundation (SNSF) and the Slovenian Research And Innovation Agency (ARIS) as a part of the WEAVE framework Grant Number 213148 (ARIS project N1-0290). R.V., A.M. and D.M. thank ARIS for funding the research program P1-0040, and D.M. thanks ARIS for funding the research program J7-3146. D.M. acknowledges funding from the European Research Council, within Grant Agreement 101141410 (HIMMS). B.L. and M.T. acknowledge funding from the ARIS (Research Core funding No. P2-0415) and the University of Ljubljana (project under Contract No. SN-ZRD/22-27/0510). C.B., W.H., A.G.M. and G.A. acknowledge funding from the European Research Council under the European Union's Horizon 2020 research and innovation program, within Grant Agreement 810451 (HERO). N.H. received funding from the European Union's Horizon 2020 research and innovation program under the Marie Sklodowska-Curie Grant Agreement 884104 (PSI-FELLOW-III-3i).

## Author contributions

C.B., D.M., and S.G. conceived the project with input from G.A. and Y.E. C.B. and D.K. fabricated the device with input from D.S. C.B., A.M., N.H., S.-W.H., F.D., and S.G. characterized the device. C.B., N.H., D.F.S., R.V., F.D., D.G., and S.G. prepared the experiment. C.B., N.H., D.F.S., W.H., H.G.B., R.V., F.D., S.-W.H., A.G.M., and S.G. carried out the experiment with input from D.G. C.B., W.H., H.G.B., and N.H. analyzed the data. B.L. conducted the simulations with input from M.T., C.B., A.M., D.M., and S.G. C.B., N.H., D.M., and S.G. wrote the manuscript with input from all co-authors.

## Competing interests

The authors declare no competing interests.
