## [Transparent Peer Review file · Nature Communications]

Imaging of electrically controlled van der Waals layer stacking in 1T-TaS₂

Corresponding Author: Dr Simon Gerber

Version 0:

Reviewer comments:

Reviewer #1

(Remarks to the Author)

The manuscript "Imaging of electrically controlled van der Waals layer stacking in 1T-TaS₂" describes an operando micro-beam x-ray study of cryogenic electrical switching of the charge density wave state in TaS₂, namely into the purported "hidden state." By correlating electrical resistance with x-ray diffraction maps, the authors identify both the spatial location and structural signatures of the electrical pulse induced hidden state in a lateral electrode geometry on 500 nm thick TaS₂. They find that the lateral and out-of-plane ordering in the electrically induced hidden state well matches that of the optically induced one reported in a previous study. In addition, they produce 2D maps which illustrate the location of the formed hidden state, that it is not limited to the surface but extends into the bulk of the crystal and beneath the electrodes. Charge injection and strain are proposed to have roles in the observed behavior. These findings are important for understanding the mechanism of non-thermal electrical resistance control in TaS₂ which may have applications in cryo-computing.

The experiments are interesting and of good quality, the results are significant, and the manuscript is clearly written. As such, I believe the work may be of sufficient interest for publication in Nature Communications. However, there are some questions that I feel need to be addressed in a revision.

1. In Fig 2, why do the lattice peak maps have so much more spatial variation than the CCDW peak maps? Please discuss the origin of the spatial variation of lattice peak signal in the text.
2. In Fig 2, why do the signal levels in lattice, CCDW, and HCDW maps drop in the C state? Is this a sign of degradation of the material? Please explain the signal drop when switching to state C in the text.
3. In the methods section, the thickness of the TaS₂ flake is stated to be 500 nm. How was this measured? Is the thickness uniform across the flake? Please explain how the thickness was measured in the text and discuss the thickness variation if measured.
4. Related to the previous point, it is argued that there is only intensity increase in the CCDW but not the lattice peak at the lower left edge of the flake in Fig 2. However, considering the spatial variation in the lattice peak signal, it seems hard to tell by eye whether the lattice peak increases or not, and I can pick out increased intensity in the lattice signal at the bottom left edge compared to the upper right. Please compute a line profile of the lattice and CCDW peak signals within the electrode gap along the length of the electrodes and compare them to confirm that there is not a commensurate increase in the lattice peak signal. If there is an increase in lattice signal, then please note that such a thickness increase could explain the apparent enhanced CCDW signal and could also contribute to preference for current flow and hence HCDW formation in that region.
5. It is argued that the switched area may have extended beneath the electrodes due to strain effects. However, in the finite element method simulations, a conductivity anisotropy only up to 5 is simulated. In D. Svetin et al. Scientific Reports (2017) doi:10.1038/srep46048, a resistivity ratio on the order of 1000 is measured between out of plane and in plane directions. Why is only a factor up to 5 simulated? If the out-of-plane resistivity is in fact much higher than the in-plane, it could help explain why the switched region extends beneath the electrodes. Considering that the deposited electrodes will coat the sides of the flake, if in-plane conductivity is much higher than out-of-plane conductivity then it could be feasible that current flows from one edge of the flake to the other, rather than just between the edges of the contacts. Considering this, please simulate the Joule heating for an anisotropy factor of 1000 and include the results or provide clear justification for why this is not done. If such high anisotropy factor can explain the extension of switched region beneath the electrodes, then please modify the discussion accordingly. For instance, it is currently stated in the text that "we identify strain propagation via the CDW domains as an essential factor," however the developed out-of-plane strain could be merely a byproduct of the transition in this case.
6. Related to the previous point, if the current flows between the flake edges rather than between the electrodes, then the

asymmetric shape of the flake could explain the asymmetric shape of the switched region. If high resistivity anisotropy can also explain asymmetric switching in the intergap space, then please modify the discussion accordingly.

7. It is discussed that “the switching channel does not seem to form along defects or impurities” due to the size of the switched region. This statement seems a bit confusing as it could be that the switched region initially nucleates at defects (e.g. the flake edge, dislocations, the electrode/flake interface, etc.) and then grows across the flake, but the measurements here do not have sufficient spatial resolution or temporal information to determine this. Please clarify this discussion or consider removing it.

8. The article refers to “partially” and “fully” switched states. However, is the state switched with 105 mA truly “fully” switched? It seems that more of the CCDW could be converted to HCDW with higher current pulses since only a small fraction is thus far switched. If so, then the nomenclature of “fully switched” is a bit confusing and I suggest the authors clarify in the text that “fully switched” here refers merely to the amount of switching done in this experiment, not the maximum amount of switching that could be achieved. On the other hand, if this is in fact as much as this device can be switched, I suggest discussing what limits more of the CCDW from being switched.

9. In Fig 1f:

a. In the current arrangement, the x axis labels in the top and bottom panels look at first glance like they are related, but in fact the top axis label only refers to the top panel. I would suggest moving the x axis in the top panel to the bottom of that top panel and then removing the space between the lower two panels to make it clearer that these x axes should be treated separately.

b. (d) in the x-axis label Time (d) presumably represents days? Please define this in the figure caption.

c. Why do the current pulses in the bottom panel appear to have amplitudes that are not monotonically increasing, especially when switching to C? i.e. sometimes later pulses have less current than earlier pulses. Please explain the pulse procedure/behavior in more detail in the text or the supplemental material.

10. In the conclusions, it is stated “ultimately, we envision separate electrode pairs that allow for multiple switched regions...” Is this the electrode arrangement proposed in A Mraz et al, Nano Lett 2022

<https://doi.org/10.1021/acs.nanolett.2c01116>? If so, then this work and any others using it should be cited here. If it is a different idea, please clarify a bit what is being proposed and cite any relevant work.

11. In the methods, the incident x-ray energy range is listed as scanning between 9.1 and 12.0 keV. Please provide the step size in energy as well as the exposure time per frame.

Reviewer #2

(Remarks to the Author)

This manuscript reports an interesting effort by using 3D tomography to image the electrical phase change behavior in a 1T-TaS₂ crystal. Advanced micron-sized X-ray light sources were employed to analyze the location and depth profiles of the hidden CDW regions from a pristine CCDW sample. The origin of the hidden phase is attributed to charge flow and lattice strain. However, the manuscript remains immature to be published at this stage. The main flaw lies in the novelty and eye-catching information. Also, the reviewer is not convinced by the experimental design and discussion and, hence, would like to express the concerns as below.

1. It is not safe for the authors to rule out the origin of filamentary conduction paths simply from the uniform imaging on the distribution of the hidden phase region (Figure 2i). Since the size of the X-ray beam is $1.5 \times 2.5 \mu\text{m}^2$, it is impossible to resolve the tiny filamentary conduction paths, if there are in the size of nanometers. In a word, the “big” micron-sized X-ray facility is inappropriate to detect such kind of “tiny” things; the experimental design itself is illogic.

2. What is the real role of lattice stain? Is it the cause for or the consequence after the phase change? In my opinion, it is also not safe to draw a conclusion from the experimental data presented in the manuscript. More control experiments or, at least, conserved discussion is required.

3. What is the nature of the phase change in the manuscript? Electronic or thermal? It seems that the authors prefer to adopt the former. However, from the bottom panel of Figure 1f, it seems that continuous long current durations are used to stimulate the phase change because the unit of time is day (d) in the horizontal axis. The use of such long pulse durations is inconsistent with literature (e.g. Ref 32), which states that only short electrical pulses less than 600 ms works for the phase change by saying “write was demonstrated with electrical pulse lengths ranging from 16 ps to 600ms”. Thus, the reviewer is wondering whether the phase change behavior presented in this manuscript is exactly same to those shown in literature. Does it a new phenomenon, or one with a different physical origin?

4. Many key information is missing, such as the parameters to control the phase change. What is the real trigger for the phase change? Is it the current applied, local electric field between electrodes, or even the Joule heat in the flake? They are intertwined and deserves a clarification. And what is the threshold value of it?

5. As usual, what is the reproducibility of the result? Can the author repeat the observation by applying the current with a same magnitude on the same sample with slightly misaligned electrodes? It is recommended to present it in the Supporting Information.

Reviewer #3

(Remarks to the Author)

This paper investigates the electrically induced charge-density wave (CDW) phase transition in 1T-TaS₂ using various in

operando micro-scale analytical techniques. By comparing the electrically induced hidden CDW phase (e-HCDW) with the previously well-studied optically induced hidden CDW phase (o-HCDW), the study demonstrates that the two phases are structurally and electronically identical. The research is systematically conducted using advanced analytical methods, providing valuable insights into electrically driven phase transitions, which are essential for future cryogenic memory applications.

From a fundamental research perspective, this work is significant as it enhances our understanding of electrically induced phase transitions in 1T-TaS₂. However, while the study is meaningful, it is difficult to consider it as a breakthrough or a major advancement in the field. In particular, from a cryogenic memory application standpoint, the findings lack specificity and leave several challenges unaddressed.

- Figure 1f shows that upon reheating, the resistance initially increases, remains stable for a while, and then decreases back to the equilibrium high-temperature state. The paper does not provide a clear explanation of this intermediate transition state. Further discussion is needed to understand its origin and implications.
- The study states that the phase transition begins at the edge of the TaS₂ flake where current injection occurs. However, the reason why the transition starts predominantly at the lower-left region is not explained. Additional discussion or experimental validation is necessary to clarify this point.
- Given that the metal electrode has a low resistance, current injection is expected to occur primarily at the boundary between the metal electrode and the channel, leading to current crowding. This should result in phase transition initiation along that boundary. However, the paper does not sufficiently explain why the phase transition starts where it does, rather than uniformly along the electrode boundary.
- Similarly, considering current crowding effects, it is questionable whether the boundary region, rather than the area beneath the metal electrode, should be the dominant site of phase transition initiation. A more thorough discussion of this point is necessary.
- The accuracy of the depth-dependent phase distribution analysis presented in Figure 4 should be discussed in more detail. Additional supporting experiments, such as ex-situ structural analysis using TEM, could strengthen the conclusions.
- The study does not provide a detailed discussion on how the observed phase transition characteristics vary with device thickness and channel length. Understanding these dependencies is crucial for both fundamental insights and potential applications.
- The introduction mentions the potential for cryogenic memory applications. However, from a memory perspective, the observed characteristics present several limitations: 1) The ON/OFF ratio is relatively low. 2) The OFF-state current is high. 3) The phase transition speed is quite slow. 3) A large current is required to induce the phase transition. Given these limitations, the paper should provide a more thorough discussion of the practicality and feasibility of using this mechanism for memory applications.

Version 1:

Reviewer comments:

Reviewer #1

(Remarks to the Author)

The authors have made significant efforts to address my comments and I feel most of them are now addressed. One issue remains regarding my previous comment #4: This was not meant to ask about intensity increase from unswitched to switched, but rather I was referring to the authors' statement in Discussion paragraph 5 that "Interestingly, the original CCDW intensity is higher in the region where the switching channel forms. Observed only for the CCDW but not the lattice peak (Fig. 2a, b), this indicates that structural differences, such as variations in flake thickness, are not responsible but rather ... potentially promoted by a particularly coherent (well-ordered) CDW state."

I agree that a region of enhanced CDW ordering could be a possible explanation, however it does also look like the lattice peak signal increases in Fig. 2a from the upper edge to the lower edge within the channel region (i.e. from yellow at upper right to brown at lower left)

Granted, the lattice peak signal is more variable for experimental reasons the authors have now clarified, but nonetheless by taking a line profile of the lattice peak signal from the upper to lower edge (panel 2a) and doing the same for the CCDW signal (panel 2b) it seems the authors will find a correlation, which may be a thickness variation, and that this possibility likewise needs to be acknowledged in Discussion paragraph 5, rather than ruled out as it is currently written. It would be fair for the authors to discuss why they do not think it is thickness variation, but I mainly think they should not claim that this intensity gradient is "observed only for the CCDW but not the lattice peak" without more careful consideration, considering that it is not obvious from visual inspection that this is really the case.

After clarifying this point I think this manuscript is suitable for publication in Nature Communications.

(Remarks on code availability)

Reviewer #2

(Remarks to the Author)

I appreciate for the author's efforts in clarifying on my concerns. With my concerns #1 and #3 being cleared, however, I feel that the concerns Nos. 2, 4 and 5 require further efforts.

For question #2, there is no evidence to show the role of out-of-plane lattice contraction, which is believed by the authors to facilitate the phase change. Why the sole charge flow mechanism is not sufficient to trigger phase change. In my understanding, the strain would emerge at the boundaries when phase change occurs and lattice constants change. It is just the consequence, rather than the cause for the phase change. More evident characterization is needed to show the role of the authors insisted on the statement "...collective growth of the hidden phase is driven by a combination of charge flow and lattice strain." Each statement should be logically accurate.

Just as the concern #7 raised by the 1st reviewer, I am also quite confused about the "partial" switched states. For question #4, it would be appreciated if the authors can provide a plot of resistance versus applied current. With it, we can know the threshold current to trigger the partial or full phase change, which in my opinion is important to understand the status of the device.

It is a rare case that the data from just one device can pass the screening of reviewer for Nat. Commun. Actually, I have the same feeling with the concern #5 from the 1st reviewer; the electrode at the left bottom of the sample is quite unusual that is rich of edge and leave many ambiguous points. For instance, the contact resistance is reduced and local electric field is enhanced, resulting in the occurrence of phase change. After reading the manuscript, I am still confused with and doubt on the origin of phase change? What is the geometry: uniform or filamentary? For question #5, I would appreciate if the authors can reproduce the experiment with an irregular TaS₂ flake and electrode definition in its location. In such a case, the phase change may occur at any random location, instead of from the bottom.

(Remarks on code availability)

Reviewer #3

(Remarks to the Author)

The authors appear to have adequately addressed the scientific issues and questions raised in the previous review, and the experimental limitations have been reasonably acknowledged. Although concerns remain regarding the proposed cryogenic memory application, the manuscript is considered sufficiently improved to warrant publication.

(Remarks on code availability)

Version 2:

Reviewer comments:

Reviewer #1

(Remarks to the Author)

The authors have now adequately addressed all of my comments and I would recommend publication in Nature Communications.

(Remarks on code availability)

Reviewer #2

(Remarks to the Author)

The authors have addressed all my concerns with additional explanation and made their points clearer in the revised version. The reviewer thinks that this paper provides useful information for the research community and can be published without further revisions.

(Remarks on code availability)

Response to Reviewer's comments

We thank the three Reviewers for their careful assessment of our manuscript and their valuable feedback. Below, we present a detailed, point-by-point response to their comments and suggestions. With these revisions, along with the updates to the manuscript and Supplementary Information, we trust to have addressed all concerns and hope that the manuscript is now ready for publication in *Nature Communications*.

Response to Reviewer 1

Reviewer 1 wrote:

*The manuscript "Imaging of electrically controlled van der Waals layer stacking in 1T-TaS₂" describes an operando micro-beam x-ray study of cryogenic electrical switching of the charge density wave state in TaS₂, namely into the purported "hidden state." By correlating electrical resistance with x-ray diffraction maps, the authors identify both the spatial location and structural signatures of the electrical pulse induced hidden state in a lateral electrode geometry on 500 nm thick TaS₂. They find that the lateral and out-of-plane ordering in the electrically induced hidden state well matches that of the optically induced one reported in a previous study. In addition, they produce 2D maps which illustrate the location of the formed hidden state, that it is not limited to the surface but extends into the bulk of the crystal and beneath the electrodes. Charge injection and strain are proposed to have roles in the observed behavior. These findings are important for understanding the mechanism of non-thermal electrical resistance control in TaS₂ which may have applications in cryo-computing. The experiments are interesting and of good quality, the results are significant, and the manuscript is clearly written. As such, I believe the work may be of sufficient interest for publication in *Nature Communications*. However, there are some questions that I feel need to be addressed in a revision.*

Our response:

We thank Reviewer 1 for acknowledging the quality of our work and supporting publication in *Nature Communications* following this round of revision.

1. In Fig 2, why do the lattice peak maps have so much more spatial variation than the CCDW peak maps? Please discuss the origin of the spatial variation of lattice peak signal in the text.

Lattice Bragg peaks are much sharper than the CCDW peaks that are elongated especially along the out-of-plane direction. Using our 3D reciprocal space mapping approach, the resolution is limited by the energy step size. In other words, there is less sampling of the much sharper lattice peaks that results in higher sensitivity or an amplification of local variations. To quantify the associated randomness of the 2D intensity images in the unswitched state (see Fig. 2a,b), we calculate the standard deviation divided by mean intensity per pixel. For the lattice and CCDW peaks, we obtain a randomness value of 0.23 and 0.18 (a.u.), respectively. Thus, while the observed spatial variations are higher in the lattice compared to the CCDW image, the roughness is of the same order.

We have added a sentence in the results section (p. 3, 2nd paragraph) explaining the origin of these fluctuations: "...There is more spatial variation in the lattice than the CCDW images, since the latter Bragg peak is broader (particularly along the out-of-plane direction) and the conversion to 3D reciprocal space is limited by the energy step size."

2. In Fig 2, why do the signal levels in lattice, CCDW, and HCDW maps drop in the C state? Is this a sign of degradation of the material? Please explain the signal drop when switching to state C in the text.

We thank Reviewer 1 for this comment, which allowed us to improve the systematics of our data analysis. Namely, we have carefully rechecked how the intensities in Fig. 2 were obtained in the submitted manuscript and noticed that originally the sum of the collected intensities was displayed. This resulted in an apparent drop in intensity upon switching because a different number of energies (frames) was measured for the different states.

In the resubmitted version of the manuscript, we have adapted Fig. 2 to display the more appropriate *mean* of the intensities, not the sum. After this correction, the apparent intensity drop upon switching is not observed. In addition, we have corrected the labelling of the lattice peak maps, Fig. 2 and Tab. 1, to (014) lattice [instead of (013) lattice].

We have also adapted Fig. S4, S5 and S7 in the Supplementary Information, accordingly, including the captions, as well as a note in paragraph C.

3. In the methods section, the thickness of the TaS₂ flake is stated to be 500 nm. How was this measured? Is the thickness uniform across the flake? Please explain how the thickness was measured in the text and discuss the thickness variation if measured.

The thickness of the flake was measured using a profilometer (model: Veeco Dektak 8 Advanced Development Profilometer), resulting in an accuracy of about ± 10 nm. We did not notice a thickness variation across the flake.

To clarify this, we have added a respective sentence in the first paragraph of the methods section (p.4): "... determined using an optical microscope and a profilometer, respectively."

4. Related to the previous point, it is argued that there is only intensity increase in the CCDW but not the lattice peak at the lower left edge of the flake in Fig 2. However, considering the spatial variation in the lattice peak signal, it seems hard to tell by eye whether the lattice peak increases or not, and I can pick out increased intensity in the lattice signal at the bottom left edge compared to the upper right. Please compute a line profile of the lattice and CCDW peak signals within the electrode gap along the length of the electrodes and compare them to confirm that there is not a commensurate increase in the lattice peak signal. If there is an increase in lattice signal, then please note that such a thickness increase could explain the apparent enhanced CCDW signal and could also contribute to preference for current flow and hence HCDW formation in that region.

To quantitatively assess intensity changes of the lattice, CCDW and HCDW signals across the flake, instead of only comparing a line trace within the electrode gap as suggested by Reviewer 1, we use the full information of our 2D maps by dividing the partially- and fully-switched mean intensity maps by the unswitched one.

In the figure on the right, the first column displays the division of the partially-switched and the unswitched state (**B/A**), whereas the second column shows the ratio of the fully-switched and unswitched state (**C/A**). The first, second and

third row show these ratios for the lattice, CCDW, and HCDW peak, respectively. The electrode gaps are outlined with pink dashed lines, and we only show the flake area for better visibility.

In these lattice maps we can see that there is no increase in intensity upon switching. In contrast for the CCDW peak—as expected—there is reduced intensity in the region where the hidden state appears upon pulsing. The HCDW peak shows the opposite behavior of the CCDW peak, *i.e.* a HCDW peak appears upon pulsing.

5. It is argued that the switched area may have extended beneath the electrodes due to strain effects. However, in the finite element method simulations, a conductivity anisotropy only up to 5 is simulated. In D. Svetin et al. Scientific Reports (2017) doi:10.1038/srep46048, a resistivity ratio on the order of 1000 is measured between out of plane and in plane directions. Why is only a factor up to 5 simulated? If the out-of-plane resistivity is in fact much higher than the in-plane, it could help explain why the switched region extends beneath the electrodes. Considering that the deposited electrodes will coat the sides of the flake, if in-plane conductivity is much higher than out-of-plane conductivity then it could be feasible that current flows from one edge of the flake to the other, rather than just between the edges of the contacts. Considering this, please simulate the Joule heating for an anisotropy factor of 1000 and include the results or provide clear justification for why this is not done. If such high anisotropy factor can explain the extension of switched region beneath the electrodes, then please modify the discussion accordingly. For instance, it is currently stated in the text that “we identify strain propagation via the CDW domains as an essential factor,” however the developed out-of-plane strain could be merely a byproduct of the transition in this case.

We agree with Reviewer 1 that it is instructive to also conduct FEM simulations assuming higher conductivity anisotropy values. Following the method described in the Supplementary Information (considering only the outer electrodes and a reduced electrode gap on one side of the flake), we performed such additional simulations with a conductivity anisotropy factor of 1000 and an applied current of $I = 9$ mA. The charge injection threshold was kept at 5×10^8 A/m².

The results of the simulation with the above parameters are presented in the left-most figure below. It can be observed that the device still preferentially switches at the side where the contacts are closer together, although a narrow channel of switched domains now also appears at the opposite edge which, however, is inconsistent with the experimental results. Interestingly, there are indeed signs of switching also beneath the electrodes, albeit not nearly as extensive as found in the experiment.

To investigate the possible influence of electrode side-coating, as raised by Reviewer 1, we performed additional simulations with separated electrode-flake contact geometries. In the middle figure above, we assumed ideal contact only from the top surface, while the vertical sides of the flake were supposed to be non-conductive. The switching remains at the narrower side in this case yet is laterally more spread out than in the experiment, and the switched material is found in the top layers of the flake. In the right-most figure, we assumed the opposite case, with only the vertical edges of the flake conductive and the top surface of the flake non-conductive. Here it can be observed that switching indeed appears beneath the electrodes—consistent with Reviewer 1’s assumption—though the protrusion of the switched material beneath the

electrodes is not nearly as extensive as indicated by the experimental results, even in this extreme simulation case with the top surface fully disconnected from the electrode. In addition, switching also occurs on both sides of the electrode channel.

In reality, however, we expect significant contact both from the top and the sides, given the 160 nm thickness of the Au electrodes and their homogeneous coverage over the flake's top and side surfaces. We agree that initial switching beneath the electrodes may result from such side coverage and could involve several flake layers which can lead to additional switching under the electrodes, as Reviewer 1 points out. Whilst it seems that the anisotropy and the electrode geometry are important for the defining the switching location and extent, none of our additional simulations reproduced the experimentally observed switching, which extends much farther beneath the electrodes. We therefore maintain our interpretation that the electrode geometry, primarily defines the switching location, and that long-range effects such as strain—also independently measured—play a key role in the switching mechanism.

We have added the figure and explanations above to the Supplementary Information.

6. Related to the previous point, if the current flows between the flake edges rather than between the electrodes, then the asymmetric shape of the flake could explain the asymmetric shape of the switched region. If high resistivity anisotropy can also explain asymmetric switching in the intergap space, then please modify the discussion accordingly.

We agree with Reviewer 1 that we cannot exclude that imperfections, such as the asymmetric shape of the flake, influence the switching location. Though, we consider this to be a second-order effect compared to the geometry of the electrodes which is well reproduced in the FEM simulations.

But to acknowledge this possibility, we have added a sentence in the discussion section (p. 4, 1st paragraph): "... though the irregular shape of the flake could also favor a switching location."

7. It is discussed that "the switching channel does not seem to form along defects or impurities" due to the size of the switched region. This statement seems a bit confusing as it could be that the switched region initially nucleates at defects (e.g. the flake edge, dislocations, the electrode/flake interface, etc.) and then grows across the flake, but the measurements here do not have sufficient spatial resolution or temporal information to determine this. Please clarify this discussion or consider removing it.

We agree with Review 1 that impurities could nucleate the switching mechanism. Consequently, we have modified the respective sentence in the manuscript (p. 3, 2nd paragraph of the discussion section): "Although the switching channel exceeds the sizes of defects [57] or CDW domains [30] it cannot be ruled out that the switching nucleates there, ..."

8. The article refers to "partially" and "fully" switched states. However, is the state switched with 105 mA truly "fully" switched? It seems that more of the CCDW could be converted to HCDW with higher current pulses since only a small fraction is thus far switched. If so, then the nomenclature of "fully switched" is a bit confusing and I suggest the authors clarify in the text that "fully switched" here refers merely to the amount of switching done in this experiment, not the maximum amount of switching that could be achieved. On the other hand, if this is in fact as much as this device can be switched, I suggest discussing what limits more of the CCDW from being switched.

In the results section of the manuscript text, we state that "partially" and "fully" switched states are purely nomenclatures related to the resistance ratio between the unswitched and the switched states:

*“We denote the state **B** with 47% resistance compared to **A** as partially-switched, reached by applying a pulse amplitude of 63mA. Increasing the pulse height to 105mA induces another resistance drop to 27% of **A** which we refer to as fully-switched **C**.”*

As Reviewer 1 correctly points out, it is indeed possible that more of the CCDW state could be converted to HCDW and, therefore, the “fully-switched” state which we investigate here is possibly not the maximally achievable switched state of the device.

From our experience, a “well-switched” hidden state can be confirmed by measuring the resistance upon heating. If the resistance decreases initially upon heating, we identify this with a situation where an HCDW state with even lower resistance could have been induced [27], *i.e.* that not the full ON/OFF ratio was achieved. However, our resistance curve upon heating (see Fig. 1f) starts off flat which indicates a well-switched HCDW state.

9. In Fig 1f: a. In the current arrangement, the x axis labels in the top and bottom panels look at first glance like they are related, but in fact the top axis label only refers to the top panel. I would suggest moving the x axis in the top panel to the bottom of that top panel and then removing the space between the lower two panels to make it clearer that these x axes should be treated separately.

b. (d) in the x-axis label Time (d) presumably represents days? Please define this in the figure caption.

We thank Reviewer 1 for pointing out these layout optimizations of Fig. 1f with which we fully agree.

We have revised the figure accordingly and define the temporal units in the caption, which are indeed in (d) for days.

*c. Why do the current pulses in the bottom panel appear to have amplitudes that are not monotonically increasing, especially when switching to **C**? *i.e.* sometimes later pulses have less current than earlier pulses. Please explain the pulse procedure/behavior in more detail in the text or the supplemental material.*

When switching to states **B** and **C**, the current pulse amplitudes increase non-monotonically likely due to an initial partial relaxation of the device. In particular, as we apply electrical pulses, we first observe a resistance increase due to partial relaxation of the device, prompting us to reduce the pulse amplitudes—to avoid damaging the device—and gradually increase them again until the final 105 mA pulse amplitude is reached.

This procedure is now also explained in the Methods section (p. 4): “The current pulse amplitude was gradually increased until the resistance dropped. For the switching from **B** to **C**, we started with the current pulse amplitude where switching from **A** to **B** had occurred. But as we observed already during the partial switching, an intermediate increase in resistance occurred and we again applied lower pulses until the final decrease in resistance was measured.”

10. In the conclusions, it is stated “ultimately, we envision separate electrode pairs that allow for multiple switched regions...” Is this the electrode arrangement proposed in A Mraz et al, Nano Lett 2022 <https://doi.org/10.1021/acs.nanolett.2c01116>? If so, then this work and any others using it should be cited here. If it is a different idea, please clarify a bit what is being proposed and cite any relevant work.

We thank Reviewer 1 for identifying this ambiguity. Our envisioned electrode configuration differs from that in Ref. [43] and [54]. Since the hidden state region forms along the edge of the flake, we here propose using multiple electrode pairs positioned only along the flake’s edge. The flake itself

could be narrow with parallel edges and separate, non-parallel electrode pairs along both edges or even orthogonal ones on top/below the flake. An open question remains regarding potential cross-talk between electrodes on opposite edges of the flake.

In the revised manuscript we now elaborate the envisioned electrode design with more detail (p. 4, 2nd last paragraph in the results section): “Ultimately, we envision non-parallel separate electrode pairs on both flake edges, as well as orthogonal ones on top and bottom of the flake that allow for multiple switched regions and controlled cross talk between them ...”

11. In the methods, the incident x-ray energy range is listed as scanning between 9.1 and 12.0 keV. Please provide the step size in energy as well as the exposure time per frame.

To optimize the measurement strategy / reciprocal space mapping and to make the best use of the limited synchrotron beamtime available, we have chosen different energy values/steps for the measurements of the **A**, **B** and **C** states with steps ranging from 20 to 80 eV depending on the peaks appearing in the different state.

Each measurement was taken with an exposure of 100 ms per position. We note that this information is already included in the captions of Fig. 2 in the main text, as well as Fig. S1, S4, S5, and S7 in the Supplementary Information, but we have now also added a sentence in the Methods section detailing the energy step sizes (p. 5, 1st paragraph): “...incoming X-rays energy between 9.1 and 12.0 keV, with step sizes ranging between 20– 80 eV, using the undulator and monochromator. The diffraction and fluorescence signals were recorded with an integration time of 100 ms ...”

Response to Reviewer 2

Reviewer 2 wrote:

This manuscript reports an interesting effort by using 3D tomography to image the electrical phase change behavior in a 1T-TaS₂ crystal. Advanced micron-sized X-ray light sources were employed to analyze the location and depth profiles of the hidden CDW regions from a pristine CCDW sample. The origin of the hidden phase is attributed to charge flow and lattice strain. However, the manuscript remains immature to be published at this stage. The main flaw lies in the novelty and eye-catching information. Also, the reviewer is not convincing by the experimental design and discussion and, hence, would like to express the concerns as below.

Our response:

We are delighted to read that Reviewer 2 finds our work interesting and are happy to address the concerns raised below. Our manuscript reports the first X-ray microdiffraction study of a functional 1T-TaS₂ phase change device, and as such is novel and also eye-catching, in view of the detailed conclusions reached, to the broad community of scientists who are the audience for *Nature Communications*.

1. It is not safe for the authors to rule out the origin of filamentary conduction paths simply from the uniform imaging on the distribution of the hidden phase region (Figure 2i). Since the size of the X-ray beam is $1.5 \times 2.5 \mu\text{m}^2$, it is impossible to resolve the tiny filamentary conduction paths, if there are in the size of nanometers. In a word, the “big” micron-sized X-ray facility is inappropriate to detect such kind of “tiny” things; the experimental design itself is illogic.

We are happy to provide this clarification regarding the experimental design and sensitivity of our approach.

Reviewer 2 is of course correct that an X-ray beam size of $1.5 \times 2.5 \mu\text{m}^2$ cannot resolve filamentary switching paths much smaller than the beam's spatial resolution. However, what we clearly see with the micron-sized X-ray beam in our experiment is the emergence of a “large” switching region, $\approx 20 \mu\text{m}^2$ in size, with an oval extent and not a channel form even in the partially-switched state **B**. But we agree that this observation does not exclude that nm-sized filamentary channels could nucleate the switching process.

We have adapted the phrasings concerning filamentary paths in the abstract and in the discussion section (p. 3) of the manuscript to clarify this matter.

2. What is the real role of lattice stain? Is it the cause for or the consequence after the phase change? In my opinion, it is also not safe to draw a conclusion from the experimental data presented in the manuscript. More control experiments or, at least, conserved discussion is required.

In our experiment and also in Ref. [49], out-of-plane lattice contraction, and therefore strain at the boundary of the switched region, is clearly identified when comparing the CCDW and HCDW states. Measuring the structural degrees of freedom and not the electronic one, our experiment is not designed to settle the fundamental “cause or consequence” question. In the abstract and discussion of the manuscript we already discuss this circumstance

“...collective growth of the hidden phase is driven by a combination of charge flow and lattice strain.”

“...Thus, we identify strain propagation via the CDW domains as an essential factor contributing to the localized and bulk-nature of the switching, ...”

by simply establishing that strain contributes to the switching process, as our experiment has been designed to show.

3. What is the nature of the phase change in the manuscript? Electronic or thermal? It seems that the authors prefer to adopt the former. However, from the bottom panel of Figure 1f, it seems that continuous long current durations are used to stimulate the phase change because the unit of time is day (d) in the horizontal axis. The use of such long pulse durations is inconsistent with literature (e.g. Ref 32), which states that only short electrical pulses less than 600 ms works for the phase change by saying “write was demonstrated with electrical pulse lengths ranging from 16 ps to 600ms”. Thus, the reviewer is wondering whether the phase change behavior presented in this manuscript is exactly same to those shown in literature. Does it a new phenomenon, or one with a different physical origin?

We thank Reviewer 2 for pointing to this ambiguity.

Our results are perfectly consistent with the hidden state switching being a nonthermal electronic effect. In particular, all switching current pulses applied in the experiment had a pulse length of 100 μs , consistent with Ref. [32]. The key parameter that we varied during switching was only the pulse amplitude. We explain this in the manuscript:

“...To induce the e-HCDW state, single square current pulses with a width of 100 μs are applied. The pulse amplitude is increased gradually until a resistance drop occurs. ...”

Our controlled switching process, involving individual pulses followed by resistance measurements, takes time, as Reviewer 2 correctly noted from the timescale of Fig. 1f (bottom panel). We would like to emphasize two key points:

1. We never apply continuous pulses. During X-ray diffraction scans, the sample is even shorted and grounded.

2. The HCDW state observed in this experiment is non-volatile at our base temperature of 6 K, consistent with earlier reports [23-27], eliminating the need for continuous pulsing.

We note that there is indeed a discussion in the field about the non-thermal origin of the HCDW state, but this has already been carefully addressed and summarized in Ref. [27]. As outlined above, we do not find any discrepancy between the switching behavior of our device and previous reports.

To avoid this confusion, we now also explicitly state that 100 μs pulses were applied in the Methods section (“... single, square-wave pulses with pulse lengths of 100 μs were applied ...”), and that if no pulses are applied, e.g. during the X-ray measurements, the device is grounded (p. 2): “During the μXRF and μXRD measurements the device is shorted and grounded.”

4. Many key information is missing, such as the parameters to control the phase change. What is the real trigger for the phase change? Is it the current applied, local electric field between electrodes, or even the Joule heat in the flake? They are intertwined and deserves a clarification. And what is the threshold value of it?

As the answer to the earlier question shows, we do not find any evidence for a thermal switching process and our measured results align with previous reports [23-27]. In Ref. [27], the origin of the switching mechanism is extensively discussed and shows that the switching is non-thermal. Our experiment was not designed to give new insight into this particular question, nor establish optimal switching performance. Nonetheless we already provide the necessary control parameters in our texts. In particular, concerning the switching threshold, the final current pulse amplitude going from the unswitched to the partially-switched state (**A**→**B**) was 63 mA in our two-terminal setup.

5. As usual, what is the reproducibility of the result? Can the author repeat the observation by applying the current with a same magnitude on the same sample with slightly misaligned electrodes? It is recommended to present it in the Supporting Information.

We appreciate Reviewer 2’s question regarding reproducibility. Despite the harsh limitation of synchrotron beamtime, we have actually measured multiple devices in two separate measurement rounds which is quite notable. From the first to the second experiment we improved setup parameters, such as the accuracy and reproducibility of translational stages to ameliorate the spatial resolution. Importantly, during further offline electrical characterization and the two beamtimes we observed qualitatively the same behavior in all measured devices with the same device design. Extending the study to different devices with electrode designs (cf. the second-last point raised by Reviewer 1) will be the scope of future work.

We now state in the Methods section (p. 4) that qualitatively the same switching behavior was observed in two separate X-ray experiments which underlines the reproducibility of our results: “We performed two measurement rounds, observing qualitatively the same behavior across several devices.”

Response to Reviewer 3

Reviewer 3 wrote:

This paper investigates the electrically induced charge-density wave (CDW) phase transition in 1T-TaS₂ using various in operando micro-scale analytical techniques. By comparing the electrically induced hidden CDW phase (e-HCDW) with the previously well-studied optically induced hidden CDW phase (o-HCDW), the study demonstrates that the two phases are structurally and electronically identical. The research is systematically conducted using

advanced analytical methods, providing valuable insights into electrically driven phase transitions, which are essential for future cryogenic memory applications. From a fundamental research perspective, this work is significant as it enhances our understanding of electrically induced phase transitions in 1T-TaS₂. However, while the study is meaningful, it is difficult to consider it as a breakthrough or a major advancement in the field. In particular, from a cryogenic memory application standpoint, the findings lack specificity and leave several challenges unaddressed.

Our response:

We thank Reviewer 3 for pointing out the significance of our work from a fundamental research perspective, which implies that the paper represents advancement in a lively field that is worthy of publication in *Nature Communications*. In this context, it is worth noting that the focus of this experiment was indeed on the investigation of the electrical switching mechanism of the material and not the direct application as a cryomemory device which has been the focus of other, dedicated studies of our team.

- Figure 1f shows that upon reheating, the resistance initially increases, remains stable for a while, and then decreases back to the equilibrium high-temperature state. The paper does not provide a clear explanation of this intermediate transition state. Further discussion is needed to understand its origin and implications.*

First, we would like to note that we perform two-point measurements and, therefore, measure also the contact resistance which influences the resistance curves / change with temperature.

The step-like increase in resistance upon heating has been reported in many studies (using four-point measurements) and attributed to partial domain relaxation [24, 32, 43]. While we did not perform μ XRD scans during heating, this effect might be associated even with a micrometer-scale relaxation of the switched region and would be a very interesting follow-up experiment of this study.

We have added a sentence in the 1st paragraph of the results section (p.2) elaborating on behavior upon heating: "... with an intermediate resistance plateau commonly assigned to a partial relaxation of domains [24, 32], ..."

- The study states that the phase transition begins at the edge of the TaS₂ flake where current injection occurs. However, the reason why the transition starts predominantly at the lower-left region is not explained. Additional discussion or experimental validation is necessary to clarify this point.*

- Given that the metal electrode has a low resistance, current injection is expected to occur primarily at the boundary between the metal electrode and the channel, leading to current crowding. This should result in phase transition initiation along that boundary. However, the paper does not sufficiently explain why the phase transition starts where it does, rather than uniformly along the electrode boundary.*

- Similarly, considering current crowding effects, it is questionable whether the boundary region, rather than the area beneath the metal electrode, should be the dominant site of phase transition initiation. A more thorough discussion of this point is necessary.*

We appreciate Reviewer 3's follow-up questions concerning the macroscopic and microscopic effects leading to the switching location/region. We address the three questions together since they are connected.

In the discussion section, we state the primary reason for the switching location:

"... Finite element method simulations (Supplementary Information) indicate that the device geometry determines the switching channel's location. Switching occurs where the electrode gap is narrowest, providing the shortest path to ground and resulting in the highest current density during pulse application ..."

Thus, the geometry of the device seems to be dictating the switching location which is very important and beneficial knowledge for potential memory applications.

The microscopic details of the switching mechanism are not understood. Current crowding and, therefore, a preferential switching location can be one explanation. Though, our experiment is not designed to answer these questions as it measures the static outcome of the switching, and not dynamically during the switching process itself.

We have added a note in the 2nd paragraph in the discussion section (p. 3) that current crowding might also play a role for the exact location of the switching region: "..., or that current crowding effects at the interface between the electrode and the flake play a role."

• The accuracy of the depth-dependent phase distribution analysis presented in Figure 4 should be discussed in more detail. Additional supporting experiments, such as ex-situ structural analysis using TEM, could strengthen the conclusions.

The amount of unswitched CCDW layers at the bottom of the flake corresponds to an intensity ratio of 0.17 in the partially- and 0.09 in the fully-switched state, following the normalized intensity ratio notation used in Fig. 4 of the main manuscript.

We now state this explicitly in the results section (p.3): "A fraction of $\approx 17\%$ and 9% remains unswitched at the bottom of the flake between the electrodes in the partially- and fully-switched state, respectively."

Considering additional experiments, we point out that the device is still functional but that such follow-up experiments would be just *a-posteriori* at room temperature where neither the CCDW nor HCDW states are present. Of course, e.g. cryo-TEM could be used but such a measurement would still be ambiguous and depend on the details of the cooling process etc. Therefore, we conclude that such a further measurement will not add relevant information to the results already presented in the manuscript.

• The study does not provide a detailed discussion on how the observed phase transition characteristics vary with device thickness and channel length. Understanding these dependencies is crucial for both fundamental insights and potential applications.

We thank Reviewer 3 for raising this question about the practical application for memory devices. We find this direction an excellent avenue for future more device-oriented experiments, but would like to point out that that considering the constraints of a beamtime, we performed the maximally achievable investigation in terms of reproducibility and generalization.

However, in terms of looking also into different geometrical dependencies, we note that we did actually simulate different device parameters considering parallel and non-parallel electrodes, as well as disconnected, inert middle electrodes in the FEM section of the Supplementary Information.

• The introduction mentions the potential for cryogenic memory applications. However, from a memory perspective, the observed characteristics present several limitations: 1) The ON/OFF ratio is relatively low. 2) The OFF-state current is high. 3) The phase transition speed is quite slow. 3) A large current is required to induce the phase transition. Given these limitations, the paper should provide a more thorough discussion of the practicality and feasibility of using this mechanism for memory applications.

Our device had to be optimized for the complexity of combining μ XRD experiments with electrical switching and *in-situ* transport. The price to pay for these adaptations is that only two-probe measurements were possible where the contact resistance contributes and reduces the ON/OFF ratio.

Optimal device performance with smaller dimensions and using four-probe measurements has been demonstrated in Ref. [43], where it was also shown that the requirements of a (1) a high ON/OFF ratio, (2) low OFF state current and (3) exceptionally high transition speed can indeed be achieved with 1T-TaS₂ cryomemory devices.

Although focused on shedding light on the nature of the hidden state transition, our experiment helps to improve device designs by hinting at specific placement of the electrodes along the edges of the flakes, as well as using a narrow flake where the latter is also favorable from the device performance point-of-view [43].

In addition to the changes listed above, we have updated the references in both the manuscript and the Supplementary Information, along with a few other minor improvements of the text.

Response to the Reviewer's comments

We sincerely thank all three Reviewers for their thorough evaluation of our resubmission and their insightful additional feedback. Below, we provide a point-by-point response to each of their comments and suggestions. We believe that these further revisions made to the manuscript comprehensively address the concerns raised, and hope that our work is now suitable for publication in *Nature Communications*.

We followed the Editor's request to check for a 'sober' writing style and have removed phrases of primacy in the resubmitted manuscript where applicable. We also revised all text to ensure a consistent phrasing amongst the main text and the Supplementary Information. All co-authors have been asked again to link their ORCID account to the manuscript, we have updated the Data and Code Availability section and enclose an updated Editorial Checklist.

Response to Reviewer #1:

The authors have made significant efforts to address my comments and I feel most of them are now addressed.

Our response:

We thank Reviewer #1 for acknowledging our efforts in the first revision and address the remaining question below.

One issue remains regarding my previous comment #4: This was not meant to ask about intensity increase from unswitched to switched, but rather I was referring to the authors' statement in Discussion paragraph 5 that "Interestingly, the original CCDW intensity is higher in the region where the switching channel forms. Observed only for the CCDW but not the lattice peak (Fig. 2a, b), this indicates that structural differences, such as variations in flake thickness, are not responsible but rather ... potentially promoted by a particularly coherent (well-ordered) CDW state."

I agree that a region of enhanced CDW ordering could be a possible explanation, however it does also look like the lattice peak signal increases in Fig. 2a from the upper edge to the lower edge within the channel region (i.e. from yellow at upper right to brown at lower left)

Granted, the lattice peak signal is more variable for experimental reasons the authors have now clarified, but nonetheless by taking a line profile of the lattice peak signal from the upper to lower edge (panel 2a) and doing the same for the CCDW signal (panel 2b) it seems the authors will find a correlation, which may be a thickness variation, and that this possibility likewise needs to be acknowledged in Discussion paragraph 5, rather than ruled out as it is currently written. It would be fair for the authors to discuss why they do not think it is thickness variation, but I mainly think they should not claim that this intensity gradient is "observed only for the CCDW but not the lattice peak" without more careful consideration, considering that it is not obvious from visual inspection that this is really the case.

After clarifying this point I think this manuscript is suitable for publication in Nature Communications.

We are grateful to Reviewer #1 for recommending acceptance of our work, following clarification of this point.

As requested, in Fig. **a-d** we show intensity line traces in the unswitched state **A**, analyzing both the lattice (green) and the CCDW (pink) peak. The line cut is taken through the center of the electrode gap, and the resulting traces represent the intensity along the path from points 1 to 2, as labeled in the spatial maps.

Also from this presentation, we do not observe a correlation between the intensity variations of the lattice (**c**) and the CCDW (**d**) peaks. The interpretation in the manuscript of the CCDW behavior is primarily based on the presence of an intensity plateau (yellow square in Fig. **d**) which aligns with the switching region. This plateau appears only for the CCDW state but not the the lattice intensity. While Fig. **b** reveals a sharp lattice intensity spike near point 2, this feature does not appear to be correlated with the extent of the switching region.

But as Reviewer #1 also mentions, due to the higher fluctuations of the lattice peak, a direct comparison between CCDW and lattice intensity is difficult. Hence to acknowledge a possible variation in flake thickness as could be reasoned from the fluctuations of the line cut in Fig. **b**, we have adapted the 4th paragraph on page 4 from:

“Interestingly, the original CCDW intensity is higher in the region where the switching channel forms. Observed only for the CCDW but not the lattice peak (Fig. 2a,b), this indicates that structural differences, such as variations in flake thickness, are not responsible but rather that switching is potentially promoted by a particularly coherent (well-ordered) CDW state.”

to

“Interestingly, the original CCDW intensity is higher in the switching region. This is not apparent for the lattice peak which features stronger spatial variations that may obscure such correlations (Fig. 2a,b). Conversely, it may suggest that structural factors such as flake thickness are not decisive and, instead, point to switching being promoted by a particularly coherent (well-ordered) CDW state.

Response to Reviewer #2:

I appreciate for the author's efforts in clarifying on my concerns. With my concerns #1 and #3 being cleared, however, I feel that the concerns Nos. 2, 4 and 5 require further efforts.

For question #2, there is no evidence to show the role of out-of-plane lattice contraction, which is believed by the authors to facilitate the phase change. Why the sole charge flow mechanism is not sufficient to trigger phase change. In my understanding, the strain would emerge at the boundaries when phase change occurs and lattice constants change. It is just the consequence, rather than the cause for the phase change. More evident characterization is needed to show the role of if the authors insisted on the statement "...collective growth of the hidden phase is driven by a combination of charge flow and lattice strain." Each statement should be logically accurate.

Our response:

After reading this further comment, and a renewed look at the initial response of Reviewer #2, we feel that rather than a disagreement there is a misunderstanding for which we apologize. We fully agree with Reviewer #2 that the phase change is initially triggered by charge flow, considering that electrons are significantly lighter and, therefore, feature faster dynamics than nuclei. Moreover, also in our view lattice strain is a consequence rather than the cause of the switching process itself. But as our work shows, it has a long-range effect on the switching region. Our statement in the abstract, mentioned by Reviewer #2:

"...is driven by a combination of charge flow and lattice strain."

was intended to convey that lattice strain occurs *concurrently* with the electronic phase transition, not that it drives it. Also in the discussion, we emphasize this view

"...Thus, we identify strain propagation via the CDW domains as an essential factor..."

That the hidden state phase change of $1T\text{-TaS}_2$ is triggered electronically, has been measured and discussed for example in Ref. [37] and also in [Guyader *et al.* Struct. Dyn. **4**, 044020 (2017)]. To our best knowledge, the time-resolved electronic and lattice response of $1T\text{-TaS}_2$ have not been simultaneously measured to date. But given that we have conducted such combined studies earlier on other systems [Gerber, Yang *et al.*, Science **357**, 71 (2017)] these are exactly the types of challenging, follow-up experiments we are working towards. We also note that in ferroelectrics, such as BaTiO_3 , our simultaneous electrical and structural measurements have shown already decades ago Ref. [60] that charge rearrangement is responsible for a phase change and that the lattice only follows. Thus, we emphasize again that there is no disagreement with the comment from Reviewer #2.

To improve clarity and avoid this misunderstanding, we have slightly revised the abstract:

"...is driven by charge rearrangement and concomitant lattice strain."

In addition, in the paragraph before the discussion section, we now explicitly state that strain results from the electronic triggering by changing the following sentence

"... hinting that not only the out-of-plane layer reordering [29, 31, 50] but also strain and the out-of-plane conductivity [46, 53] play a role in the switching process."

to

“...hinting that not only the electronically-driven out-of-plane layer reordering [29, 31, 50], but also the resulting strain and change of the out-of-plane conductivity [46, 53] play an important role in the manifestation of the switching process.”

Just as the concern #7 raised by the 1st reviewer, I am also quite confused about the “partial” switched states. For question #4, it would be appreciated if the authors can provide a plot of resistance versus applied current. With it, we can know the threshold current to trigger the partial or full phase change, which in my option is important to understand the status of the device.

We feel that also this concern may stem from suboptimal phrasing, for which we apologise to Reviewer #2. In our view, the confusion may be due to the interpretation of the term “partial,” which can have either an electronic or spatial meaning. In our work, we use “partial” strictly in the spatial sense, *i.e.* by “partially switched” (by applying a 63 mA current pulse), we mean that only a small portion of the maximally (“fully”) switched region undergoes a phase change. But importantly this small portion switches completely from an electronic point of view. Consequently, we define the “fully switched” state as the maximally switchable portion of the flake which is connected to the maximum current pulse amplitude of 105 mA we could apply in our experiment, limited by the compliance setting of our Keithley pulsed current source. In principle, it would have been possible to apply an even higher current amplitude to potentially switch a larger spatial region of the flake. However, from our experience the available pulse amplitude in this experiment is already reasonably close to electrical breakdown of the device.

In the revised manuscript we have added a note regarding the meaning of “partial” and “full” switching (last paragraph before the discussion on page 3).

We would like to emphasize that the *IV* traces of our 1T-TaS₂ devices upon switching are generic and extensively discussed, *e.g.* in Ref. [43, 44] as well [Mraz *et al.*, arXiv:2203.14586]. But as requested by Reviewer #2, above we show two generic switching *IV* traces from two of our devices at a temperature of $T = 30$ K, though not the ones measured with X-rays – for time optimization, we did not measure this type of *IV* traces upon switching during the time-limited synchrotron beamtime. 100 μ s current pulses were applied with the pulse amplitude shown on the horizontal axis and the corresponding voltage, dependent on the resistivity of device, on the vertical axis. The drop in voltage and, therefore, resistance denotes the switching to the HCDW state. Both devices were measured in a two-probe setup. Upon applying even larger pulse amplitudes, the resistance drops further after which device destruction is risked.

It is a rare case that the data from just one device can pass the screening of reviewer for Nat. Commun. Actually, I have the same feeling with the concern #5 from the 1st reviewer; the electrode at the left bottom of the sample is quite unusual that is rich of edge and leave many ambiguous points. For instance, the contact resistance is reduced and local electric field is enhanced, resulting in the occurrence of phase change. After reading the manuscript, I am still confused with and doubt on the origin of phase change? What is the geometry: uniform or filamentary? For question #5, I would appreciate if the authors can reproduce the experiment with an irregular TaS₂ flake and electrode definition in its location. In such a case, the phase change may occur at any random location, instead of from the bottom.

We would like to reiterate – as already done in the response to the first round of reviews – that we have actually measured multiple devices across two independent synchrotron beamtimes, consistently observing qualitatively similar switching behavior. Importantly, we have confirmed the switching process not only from an electronic point of view (resistivity and CDW state), but also the lattice degree of freedom (see vanishing dimer peaks and out-of-plane lattice contraction in the Supplementary Information). Given the scarcity and high competition for synchrotron beamtime, we feel that this is already quite notable in terms of reproducibility.

Between the first and second beamtime, we have implemented significant technical improvements to the measurement setup, specifically upgrading the translational stages to accommodate the bulky and heavy ⁴He cryostat. This refinement was necessary to avoid backlash, and therefore translation jitter, to allow for spatially-resolved measurements precisely locating the HCDW state as presented in the manuscript. This level of stability is required over several days as three-dimensional reciprocal space mapping and the subsequent HCDW screening requires spatially-resolved scans at multiple energies.

As a preparation for these two synchrotron experiments, we conducted systematic electrical pre-characterization of nine devices and additionally performed structural alignment also at the synchrotron on four of these nine devices. Three out of four fully pre-characterised devices were measured during the two time-constrained micro-beam experiments. The results of the final device, the third one, is presented in the manuscript.

Given the highly competitive, externally reviewed awarding process for the limited amount of synchrotron beamtime, it is unfortunately simply not possible to repeat the measurements with further devices. But in principal, we fully agree with Reviewer #2 that performing similar experiments on regular and irregular shaped flakes with different electrode configurations is a very interesting idea for follow-up experimental campaigns, which however are beyond the scope of this proof-of-principle study.

But we include here some of the results obtained with the first two devices which qualitatively show the same switching properties, electrically and for device #1 also structurally, as device #3 presented in the manuscript.

Measurement results of device #1:

This device was measured during the initial synchrotron beamtime, using smaller, less sturdy translational stages that introduced spatial jitter, ultimately preventing us from accurately locating the switching region.

a above shows an optical image of device #1 where the gap between the outer electrodes is $8 \mu\text{m}$ and the scale bar is $30 \mu\text{m}$. **b** Electrical two-probe pre-characterisation with cooling and heating, as well as switching at low temperatures (red arrow). In this device multiple current pulses with increasing pulse amplitude up to 15 mA were applied.

a μXRF image measured at room temperature during the first beamtime to locate device #1. Outlined in yellow is the electrode gap position. **b** Spatially resolved μXRD of the CCDW peak of device #1, recorded at the base temperature of $T=6 \text{ K}$ and an X-ray energy of 9.65 keV . The electrode gap is also outlined in yellow. Note that neither the device nor the image was rotated, *i.e.* the orientation is the same as for the room-temperature μXRF data shown in **a**. The apparent slant of the device is an artifact caused by the non-ideal translational stages used to support the heavy cryostat. Signs of jitter along the horizontal scan direction are also visible and progressively worsened over the course of the initial beamtime.

The vertical summed intensity projection from the region marked by the purple square in the μXRD scan above (**b**), is shown for the unswitched CCDW state (red), as well as after applying 1 mA (blue) and 8.5 mA (green) current pulses. The position of the electrode gap is indicated in yellow. A decrease in the CCDW peak intensity around the electrode gap is observed after applying 8.5 mA , congruent with the transition to the HCDW state. Consistent behavior is seen in data taken at other X-ray energies. Unfortunately,

the mechanical jitter in this initial measurement limited our ability to pinpoint the location of the switching region. But it qualitatively agrees with the observation made with device #3

presented in the manuscript: the CCDW peak intensity decreases in the electrode gap region upon switching.

Measurement results of device #2:

This device was measured during the second synchrotron beamtime with improved translational stages, but unfortunately it was destroyed during switching process.

a Optical image of device #2 where the scale bar is 30 μm . **b** Electrical two-probe pre-characterisation with cooling and heating, as well as switching at low temperatures. Also here multiple current pulses with increasing pulse amplitude were applied, with a final one of 15 mA which destroyed the flake as is visible in the optical image, leaving it in a “bow-tie” shape.

Spatially-resolved μXRD of the lattice and CCDW peaks of device #2, measured at 10.9 keV at the base temperature of $T = 6\text{ K}$, *i.e.* before switching is shown on the right. **a** shows the raw detector image with two regions of interested (ROI) in red (lattice) and blue (CCDW). **b** and **c** depict blow-ups of the two ROIs. **d** and **e** depict the respective μXRD maps of the two signals.

Finally, we also comment on Reviewer #2’s question regarding the nature of the switching region. In our experiment, we observe a uniform switching region without hints for a filamentary conduction path – of course bearing in mind the finite spatial resolution / spot size of the measurements, as we have elaborated in the first review round. We emphasize that this observation is also supported by other studies. For example,

scanning tunneling microscopy (STM) of the e-HCDW state (Supplementary of Ref. [27]) reveals a uniform domain wall network rather than a narrow channel-like structure. Additionally, scanning tunneling spectroscopy (STS), Ref. [29], performed on both domains and domain walls in the e-HCDW state confirm two key points: (1) the switching region is uniform, consistent with the STM findings, and (2) both the domains and domain walls exhibit metallic behavior. The latter is important, as it indicates that domain walls do not serve as filamentary conduction paths. Taken together, these STM and STS results are consistent with our experimental observations and support the interpretation of a spatially extended switching region.

To emphasize this notion, in the revised manuscript we consistently use “switching region” instead of “switching channel”.

Response to Reviewer #3:

The authors appear to have adequately addressed the scientific issues and questions raised in the previous review, and the experimental limitations have been reasonably acknowledged. Although concerns remain regarding the proposed cryogenic memory application, the manuscript is considered sufficiently improved to warrant publication.

Our response:

We are pleased to read that Reviewer #3 is satisfied with our response the the first round of reviewing and recommends publication of our manuscript in *Nature Communications*. Regarding the potential of optimized 1T-TaS₂ devices for cryomemory application we refer to Ref. [32, 43], as well as our recent preprint [Mraz *et al.*, arXiv:2203.14586].